# An empirical comparison of some missing data treatments in PLS-SEM

**Lateef Babatunde Amusa**[1,2]*, **Twinomurinzi Hossana**[1]

**1** Centre for Applied Data Science, College of Business and Economics, University of Johannesburg, Johannesburg, South Africa, **2** Department of Statistics, University of Ilorin, Ilorin, Nigeria

* amusa.lb@unilorin.edu.ng

**Data Availability Statement:** All relevant data are within the manuscript and its Supporting Information files.

## Abstract

PLS-SEM is frequently used in applied studies as an excellent tool for examining causal-predictive associations of models for theory development and testing. Missing data are a common problem in empirical analysis, and PLS-SEM is no exception. A comprehensive review of the PLS-SEM literature reveals a high preference for the listwise deletion and mean imputation methods in dealing with missing values. PLS-SEM researchers often disregard strategies for addressing missing data, such as regression imputation and imputation based on the Expectation Maximization (EM) algorithm. In this study, we investigate the utility of these underutilized techniques for dealing with missing values in PLS-SEM and compare them with mean imputation and listwise deletion. Monte Carlo simulations were conducted based on two prominent social science models: the European Customer Satisfaction Index (ECSI) and the Unified Theory of Acceptance and Use of Technology (UTAUT). Our simulation experiments reveal the outperformance of the regression imputation against the other alternatives in the recovery of model parameters and precision of parameter estimates. Hence, regression imputation merit more widespread adoption for treating missing values when analyzing PLS-SEM studies.

## Introduction

Because of its predictive power combined with its explanatory strengths, partial least squares structural equation modeling (PLS-SEM) is thought to be well-suited to build and assess explanatory-predictive theories [1]. It has grown in popularity in recent years in a variety of fields, particularly marketing [2,3], information systems [4], and business and management studies [5,6].

A significant bias source in PLS-SEM is missing data. Missing data is generally a common problem in applied studies that might jeopardize the validity and reliability of research findings if not adequately addressed. The issue arises when participants have insufficient or unavailable data for one or more variables in the analysis model. This missingness can occur for various reasons, including participant non-response, attrition, or measurement error. The consequence of incomplete data can be severe, resulting in inaccurate parameter estimations, lower statistical power, and potentially incorrect conclusions.

**Funding:** The author(s) received no specific funding for this work.

**Competing interests:** The authors have declared that no competing interests exist.

PLS-SEM researchers have routinely applied listwise deletion and mean replacement to deal with missing data [7]. While more reliable results are produced by mean replacement than case-wise deletion, mean replacement artificially introduces the problem of variance reduction [8]. In general, these approaches may introduce errors that misrepresent association coefficients. Missing data is more frequently associated with respondents with some similar features. For example, some wealthy or high-income individuals may be reluctant to expose their purchasing patterns. Depressed respondents may deliberately conceal questions about anxiety. Excluding these particular groups of respondents from datasets may cause severe relationship disruptions between variables. Deletion methods limit the data points available for analysis, and fewer samples reduce the statistical technique's power.

Except for a few initiatives, such as case-wise and pairwise deletion, approaches to missing data imputation in PLS-SEM are not well-established [9]. On the other hand, quite a number of studies [8,10–12] used extensive numerical simulations to assess the utility and performance of imputation techniques in covariance-based SEM. In summary, strategies for dealing with incomplete data in PLS-SEM are rarely explored or investigated.

Filling this research gap is critical because PLS-SEM is a popular and widely utilized analysis approach. According to Hair et al. [13], the extent to which these approaches can be used to impute missing data in PLS-SEM analysis is unknown. Missing imputation approaches that rely on regression or the expectation-maximization (EM) algorithm are not generally acknowledged in the PLS-SEM literature.

This study investigates the relative performance of four missing data techniques in the context of PLS-SEM. The four methods studied were listwise deletion (or complete case analysis), mean imputation, regression imputation, and EM imputation. Specific research questions addressed how the methods comparatively (1) impact measurement model quality in terms of convergent validity, (2) introduced bias in estimating structural model parameters, and (3) affect the model accuracy as estimated by the model standard errors.

The assessments assumed three missing data mechanisms: missing completely at random (MCAR), missing at random (MAR), and missing not at random (MNAR).

## Methodology

In this study, we aim to compare different missing data techniques for Partial Least Squares Structural Equation Modeling (PLS-SEM) using Monte Carlo simulation. PLS-SEM is a widely used statistical approach for analyzing complex relationships among latent variables. Missing data is a common issue in empirical research, and it can significantly affect the accuracy and validity of the results obtained from PLS-SEM analyses. Therefore, it is important to evaluate the performance of various missing data techniques in the context of PLS-SEM to identify the most suitable approach. Though PLS-SEM frequently employs list-wise deletion and mean replacement, the classical handbook of PLS-SEM [7] acknowledged that there was limited knowledge of the applicability of regression or expectation maximization algorithm for missing data imputation.

The specific methods explored in this work are described below. They cover only a handful of the many methods presented, but we feel they reflect a spectrum of commonly used and promising approaches.

Complete Case Analysis (CCA): In CCA, also known as listwise deletion, all observations with missing values are removed from the data analysis. Case-wise deletion is simple to implement but always reduces the sample size. When nonignorable missing data is present, CCA causes further complications since the pattern of data missingness is nonrandom and not predicted from other variables in the database. This technique will only produce unbiased

estimates if the missing data are missing at random (MCAR) or if all mechanism variables have been included to make the missing data ignorable.

Mean imputation (MI): It involves replacing incomplete data with the mean of non-missing elements. This technique is popular and simple, but it reduces variable variability due to the replacement of all missing entries of the same variable by a constant value [10].

Regression imputation: Based on available data, the variable with incomplete data is regressed on other variables with complete data in a multiple regression model. The estimated outcomes of this fitted regression model are then used to replace the missing values. The variability of variables is better preserved in regression due to the variance of error terms than a constant imputation like the mean imputation.

Expectation-Maximization Algorithm (EM): A more popular approach is the use of the EM algorithm [14,15]. The EM algorithm estimates missing values by iteratively imputing values based on the available data. It starts by initializing the missing values in the dataset with reasonable initial estimates (e.g., mean imputation or random imputation). In the Expectation step, the algorithm estimates the expected values of the missing data given the observed data and the current parameter estimates. It computes the conditional means or imputation values for the missing values based on the available data and the estimated parameters. The Maximization step updates the model parameters based on the imputed data. This involves estimating the model parameters using the imputed dataset, such as estimating the latent variable scores, path coefficients, and measurement model parameters in PLS-SEM. This procedure is repeated until the parameter estimations converge.

## Missing data mechanisms

Missing data mechanisms are the many ways data can be missing or incomplete. Identifying and understanding missing data mechanisms is crucial because they could influence the validity and reliability of statistical investigations and empirical conclusions. Here are some examples of common missing data mechanisms:

Missing Completely at Random (MCAR): In MCAR, data missingness is unrelated to the observed and unobserved data. It assumes that any characteristic of the data or the data collection process does not influence the probability of data being missing. In other words, the missingness occurs randomly, and there is no systematic difference between the missing and non-missing data. Though MCAR is generally regarded as a strong and often unrealistic assumption, it is considered the ideal missing data mechanism because it allows for unbiased statistical analyses. Complete case analysis and various imputation methods, including single or multiple imputations, can handle MCAR [16].

Missing at Random (MAR): MAR arises when observed variables may explain the missingness but not the missing data itself. It then implies that after controlling for other observed variables, the probability of missingness depends solely on the observed data. Imputation methods or modeling approaches, such as the full information maximum likelihood (FIML) or expectation-maximization (EM) algorithm, can be used to address MAR [17].

Missing Not at Random (MNAR): MNAR refers to situations where the missingness is systematically related to the unobserved data. This pattern of missingness is not random and can contribute significant bias to the analysis. MNAR is more challenging; researchers frequently use sensitivity analyses or multiple imputation procedures that integrate model assumptions [16].

## Monte carlo simulation

We used Monte Carlo simulation to generate multiple datasets with known population parameters and simulate missing data under different conditions to evaluate the performance of

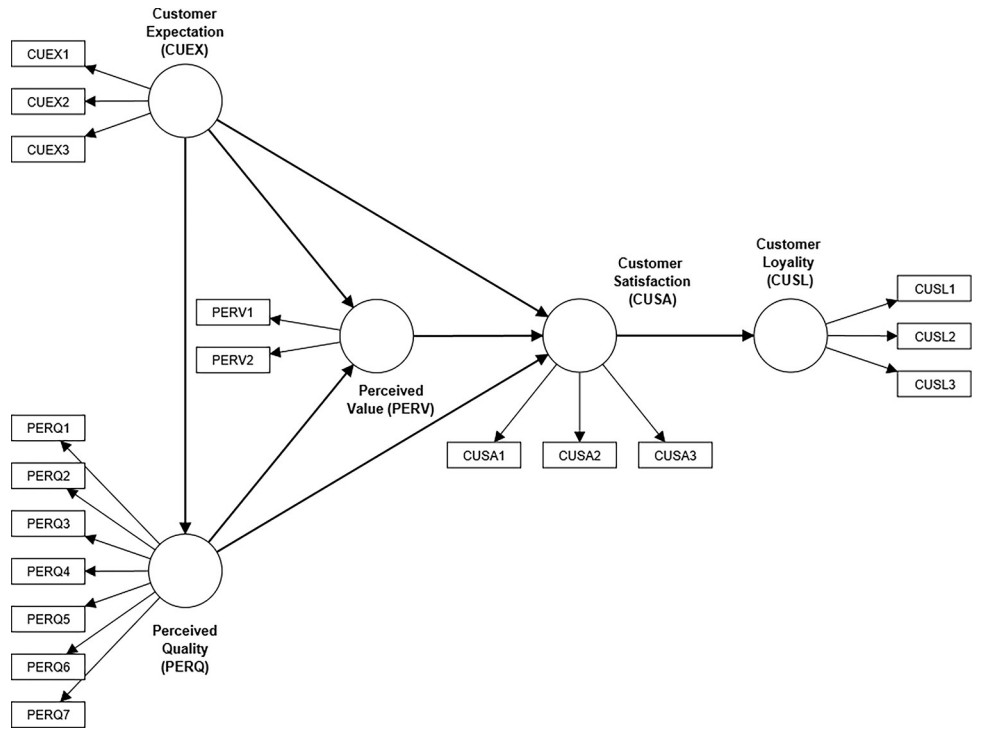

**Fig 1. Adapted ECSI model with model parameters from Tenenhaus et al.** (2005) [20].

various missing data techniques. The data will comprise latent variables and manifest indicators representing the theoretical model under investigation. The simulations are built on two theoretical models in Figs 1 and 2, which respectively mirror the structure of a customer satisfaction index model [18] and the UTAUT model [19]. We chose this simulation route because it reflects the typical complexity of structural equation models within the social science discipline.

Different missing data mechanisms, such as Missing Completely at Random (MCAR), Missing at Random (MAR), and Missing Not at Random (MNAR), were assumed to simulate missingness in the generated dataset. The PLS Mode A method with the path weighting scheme [20] was employed in the analysis.

## Design factors

The simulation study design involved two levels of sample size (300 and 1000), four proportions of missing data (20%, 30%, 40%, 50%), and three missing data mechanisms (MCAR, MAR, MNAR). These three design factors led to a 2 x 4 x 3 factorial design, with each of the 24 conditions being generated and analyzed for 500 samples, resulting in a total of 12000 analyzed samples for the methods under consideration.

## Performance measures

For each of the 12000 replications, we calculate the mean absolute error (MAE) for the structural model parameters corresponding to the theoretical models in Figs 1 and 2. The MAE is

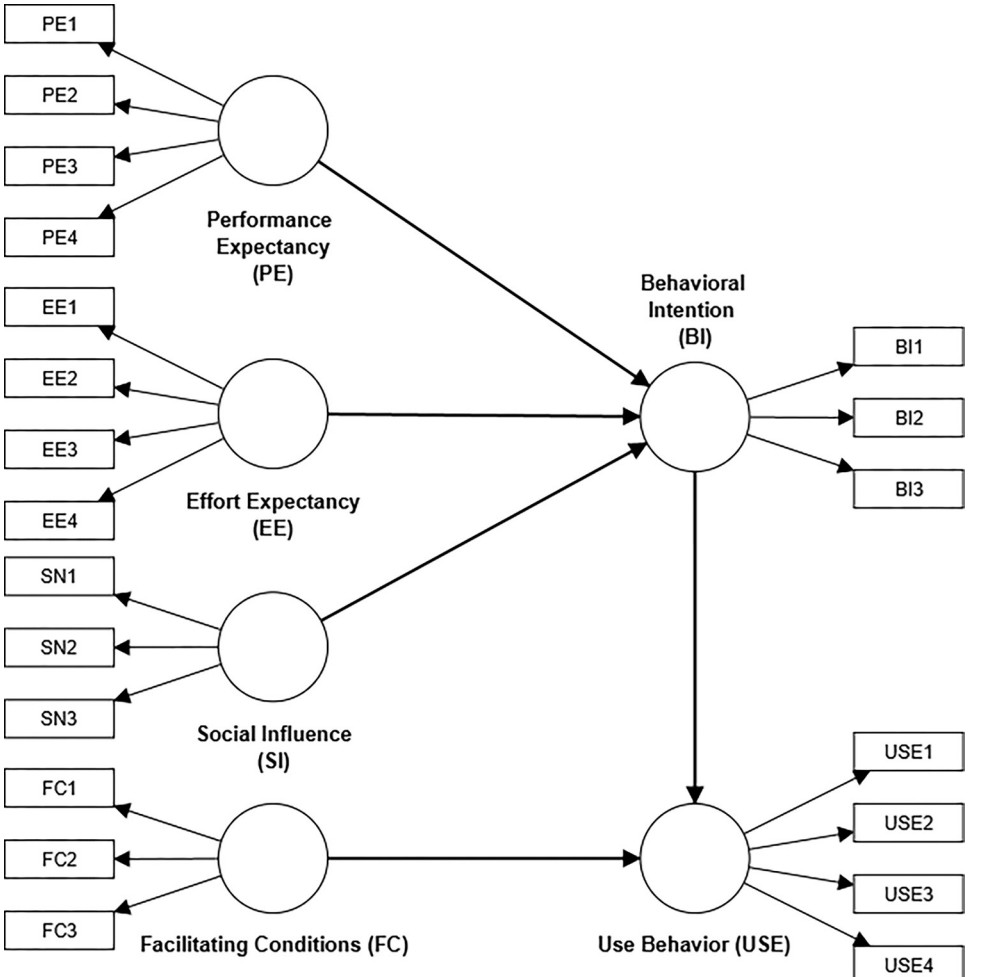

**Fig 2. Adapted UTAUT model with model parameters from Gahtani et al.** (2007) [19].

defined as

$$\mathrm{MAE} = \frac{1}{t} \sum\nolimits_{j=1}^{t} |\hat{\beta}_j - \beta_j|,$$

where $t$ is the number of parameters, $\hat{\beta}_j$ is the parameter estimate in any replication and $\beta_j$ is the prespecified model parameter.

All computations, including data generation and analyses (see **S1 File**) have been conducted within the R 4.3.1 environment [21] using the SEMinR [22] and cSEM.DGP packages [23].

## Simulated data based on the ECSI model

This simulation experiment is based on the European Customer Satisfaction Index (ECSI) [18] model and is empirically validated by a classical study [20]. The ECSI is an economic indicator developed from the Swedish customer satisfaction barometer [24]. The model is evaluated using a 10-point Likert scale for each of the five constructs provided based on well-established theories and methodologies in consumer behaviour. These constructs, which are all reflective measures, include customer expectations, perceived quality, perceived value, customer

satisfaction, customer complaints, and customer loyalty (see Fig 1). The covariance matrix of the ECSI data analyzed in a 250-sample classical study [20] was used to generate the multivariate MVs.

In terms of the measurement items, we considered equal standardized indicator loadings ($\lambda$ = 0.7) for all the constructs. This is consistent with some previous Monte Carlo studies of PLS-SEM [9,25,26] to ensure conventional acceptance thresholds of validity and reliability for the measurement models. We, however, retained the estimated path model coefficients as our prespecified path model coefficients.

### Simulated data based on the UTAUT model

In this example, we utilized a prominent unified theory of acceptance and use of technology (UTAUT) model [27] and an empirically validated study [19] as a benchmark for our Monte Carlo simulations. The adapted classical study comprises 21 indicators and six reflective constructs, each measured using a 7-point Likert scale [19]. The six constructs reflect the influences of social influence and cognitive instrumental constructs on perceived usefulness of IT and usage intentions. The constructs include performance expectancy, effort expectancy, social influence, facilitating conditions, behavioral intentions, and usage behaviour (see Fig 2). This model differs from the ones in the previous simulation example in terms of the number of manifest variables and measurement modes, and it provides another typical model for evaluating the missing data techniques.

Regarding prespecified model parameters, we retained the estimated parameter values in the adapted study for the measurement and structural model parameters. The estimated standardized loadings from the adapted study ranged from 0.71–0.95.

## Results

Results were obtained for analyses with listwise deletion, mean imputation, regression imputation, and EM algorithm. For both simulation examples, we first evaluated the influence of the missing value techniques on validity and reliability as measured by average variance extracted (AVE) and composite reliability (CR) for the measurement models (see Figs 3–5). In other words, were the conventional acceptance thresholds for validity and reliability still met after the missing data treatments?

### Simulation 1

In this simulation experiment, loading-related estimates for only one of the factor combinations are shown because all loadings are the same in the true population model. This eliminates congestion and repetition because the same pattern of results occurs with all loadings. The conventional thresholds AVE = 0.5 and CR = 0.7 were superimposed (see Fig 3).

As shown in Fig 3, the AVE and CR values were well above 0.5 and 0.7, respectively. It implies that the missing data techniques produced sufficiently high internal consistency and convergent validity for the constructs.

Regarding bias in estimating structural model parameters, results are summarized in Fig 4. Similar patterns emerged in comparing the different methods across the simulation conditions for both the small (n = 300) and large (n = 1000) sample size conditions. Regression imputation (Reg) consistently produced higher MAE values with increasing missing proportions. A similar pattern was observed for complete case analysis (CC), except for the MCAR case, where lower MAE values emerged at 30% missingness and above. An opposite pattern was observed for mean imputation (Mean), while EM produced relatively constant MAE values for all missing proportions.

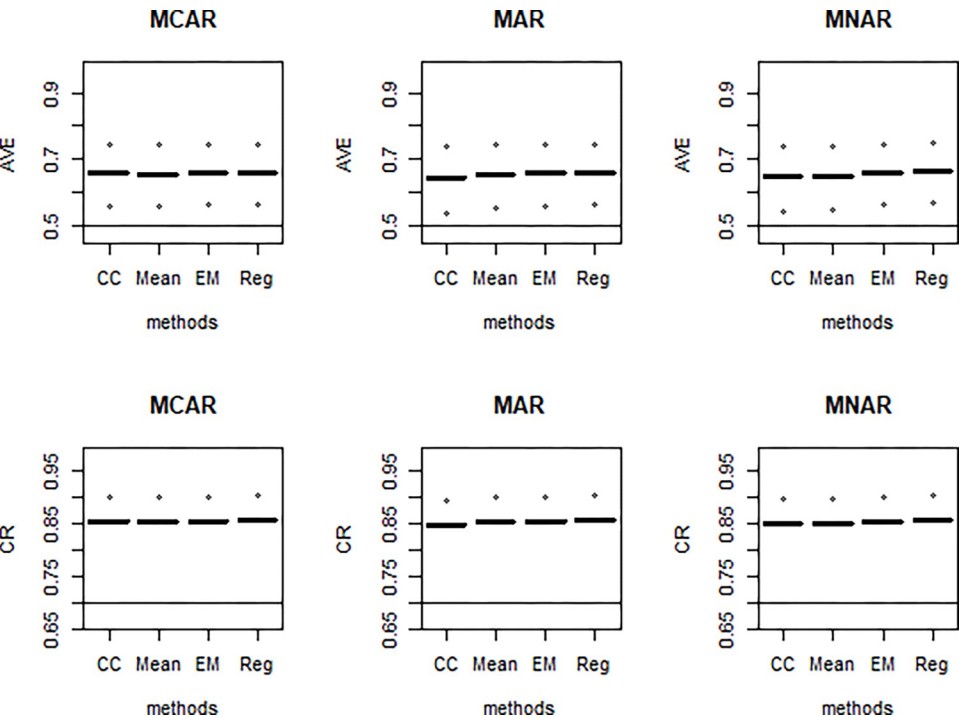

**Fig 3. AVE (Top panel) and CR (Bottom panel) of the missing data techniques under simulation model 1 for N = 300 and 20% missingness for each of the different missing mechanisms.**

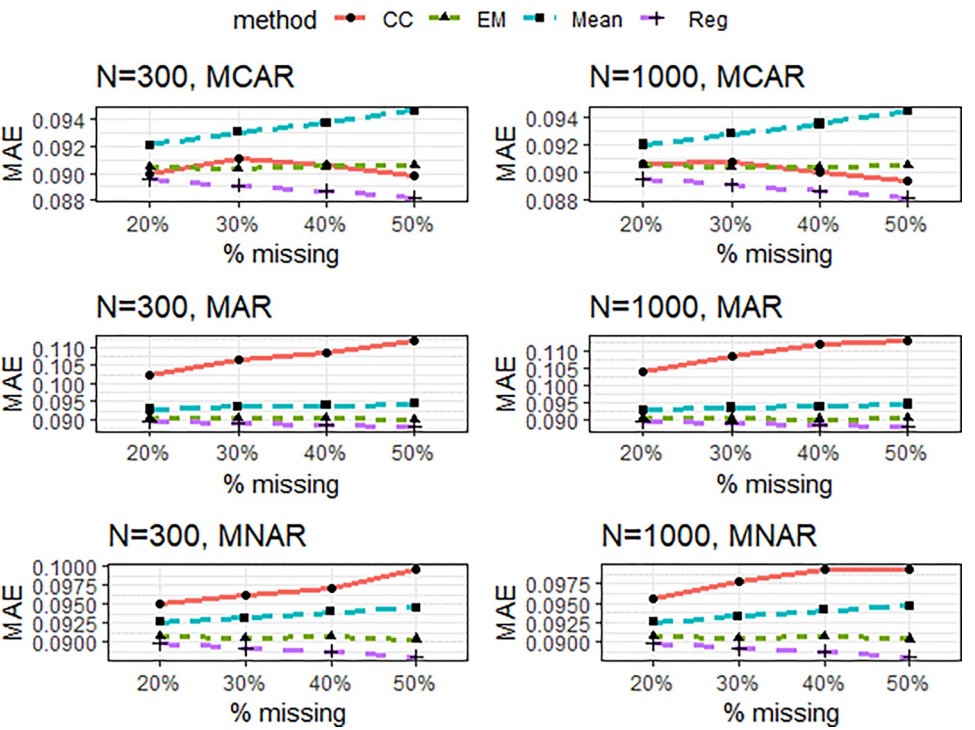

**Fig 4. Simulation example 1: Mean absolute error (MAE) of the different missing data treatments under varying design factor combinations.**

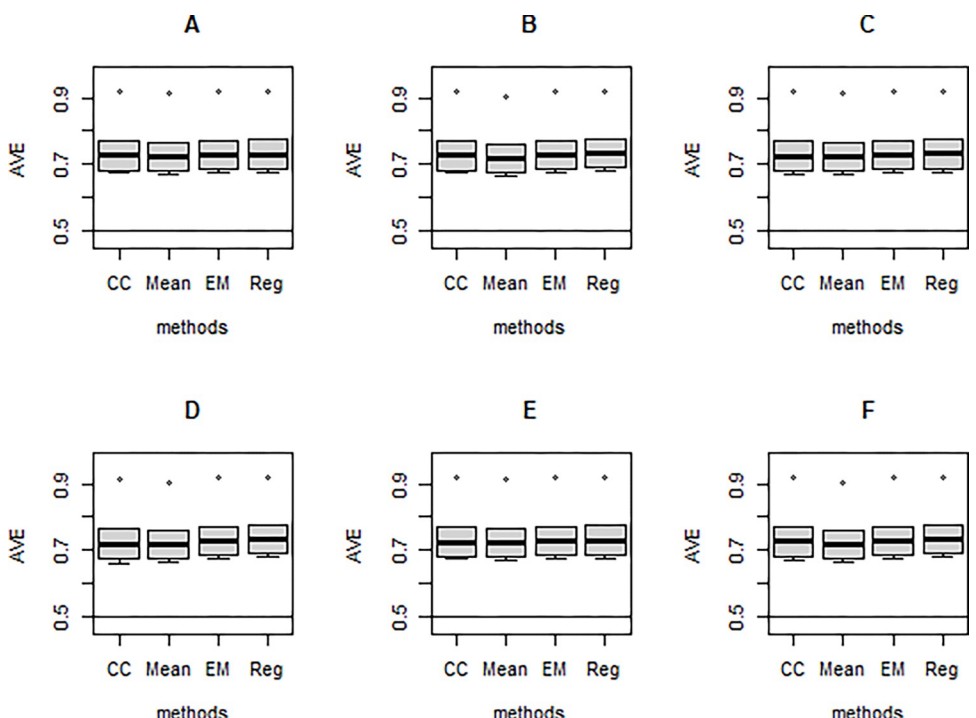

**Fig 5. AVE of the missing data techniques under simulation model 2 for some selected design factor combinations.** Panel A: MCAR, N = 300, 20% missingness, B: MCAR, N = 1000, 50% missingness, C: MAR, N = 300, 20% missingness, D: MAR, N = 1000, 50% missingness, E: MNAR, N = 300, 20% missingness, F: MNAR, N = 1000, 50% missingness.

When the missing data are MCAR, regression imputation outperformed the other methods, with EM and CC marginally competing for the 2nd spot. Similar patterns emerged for the MAR and MNAR scenarios. CC performed worst, producing substantially higher MAE values than the other methods. Regression imputation outperformed the other methods. Though in the MAR scenario, the MAE values were marginal to EM–the 2nd best, more striking differences were observed for MNAR.

## Simulation 2

For simulation example 2, conventional thresholds AVE = 0.5 and CR = 0.7 were superimposed in Figs 5 and 6, respectively. Despite unequal factor loadings in this simulation experiment, only results for some six representative factor combinations were reported due to space constraints. Besides, there was a similar pattern of results, and conclusions could be made for all 24 possible factor combinations.

Results concerning bias in estimating structural model parameters are summarized in Fig 7. Most of our findings in simulation example 1 still holds in this structural model. Overall, regression imputation still outperforms other candidate imputation methods. Although CC performed best under the MCAR scenario, the inconsistency of results under the different sample sizes is worth noting. The differences between CC and Reg appear to be marginal at the large sample size condition (n = 1000), and Reg even produced a substantially lower MAE value at the 40% missing proportion. These inconsistencies may be due to sampling fluctuations.

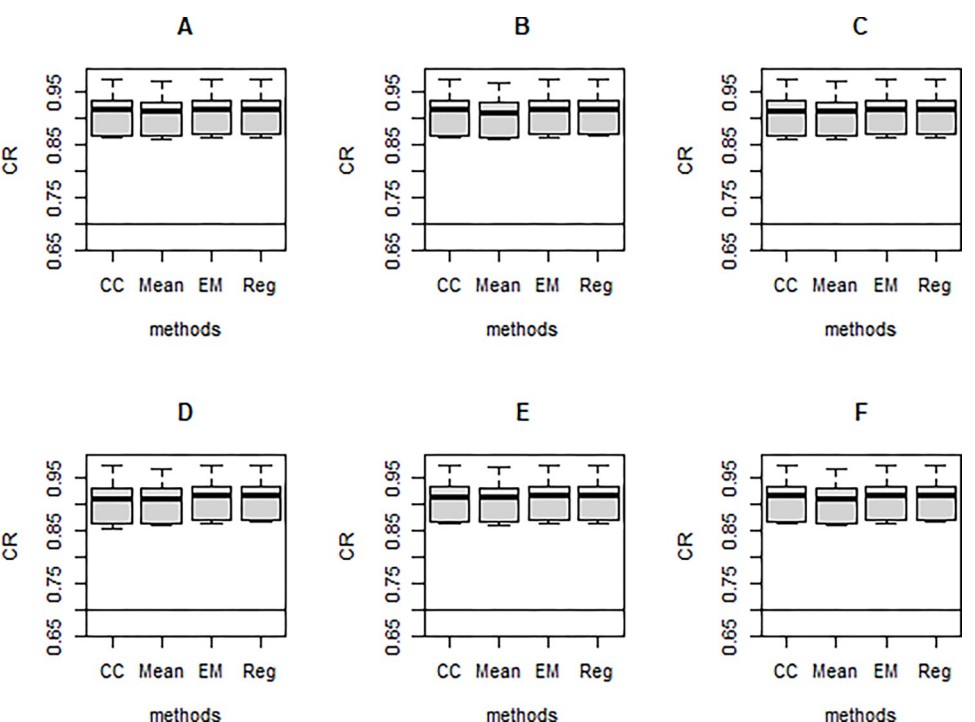

**Fig 6. CR of the missing data techniques under simulation model 2 for some selected design factor combinations.**
Panel A: MCAR, N = 300, 20% missingness, B: MCAR, N = 1000, 50% missingness, C: MAR, N = 300, 20% missingness, D: MAR, N = 1000, 50% missingness, E: MNAR, N = 300, 20% missingness, F: MNAR, N = 1000, 50% missingness.

## Precision of parameter estimates

Using the ECSI model of example 1, an additional evaluation based on a small set of design conditions was conducted to obtain the standard errors of the estimated structural model parameters. Due to the computational cost of bootstrapping in the context of Monte Carlo simulations, we limited our assessment to one sample size condition (n = 300). Other factor conditions were varied as usual. We examined precision by averaging 200 resampled bootstrap standard errors (SE) for the path coefficients over the 500 simulated datasets. The standard errors were evaluated on an aggregate level of the structural parameters.

Fig 8 provides the standard errors for the studied structural model parameters. Complete case analysis substantially produced less precise estimates. For the small and large sample size conditions, stark differences were observed between the standard errors of complete case analysis and the other candidate methods. Though unnoticeable differences and sometimes similar standard errors were observed for the other three techniques, regression imputation numerically produced more precise estimates.

## Discussion

Despite the numerous available missing data techniques, complete case analysis (or listwise deletion) and mean imputation are routinely chosen as treatments for missing data in PLS-SEM studies [28]. This is also evident in the implementation of missing data treatment available in the popular Smartpls 4.0 software [29]. We sought to evaluate other notable missing data techniques, including regression and EM imputations, and compare them with the commonly used listwise deletion and mean imputation.

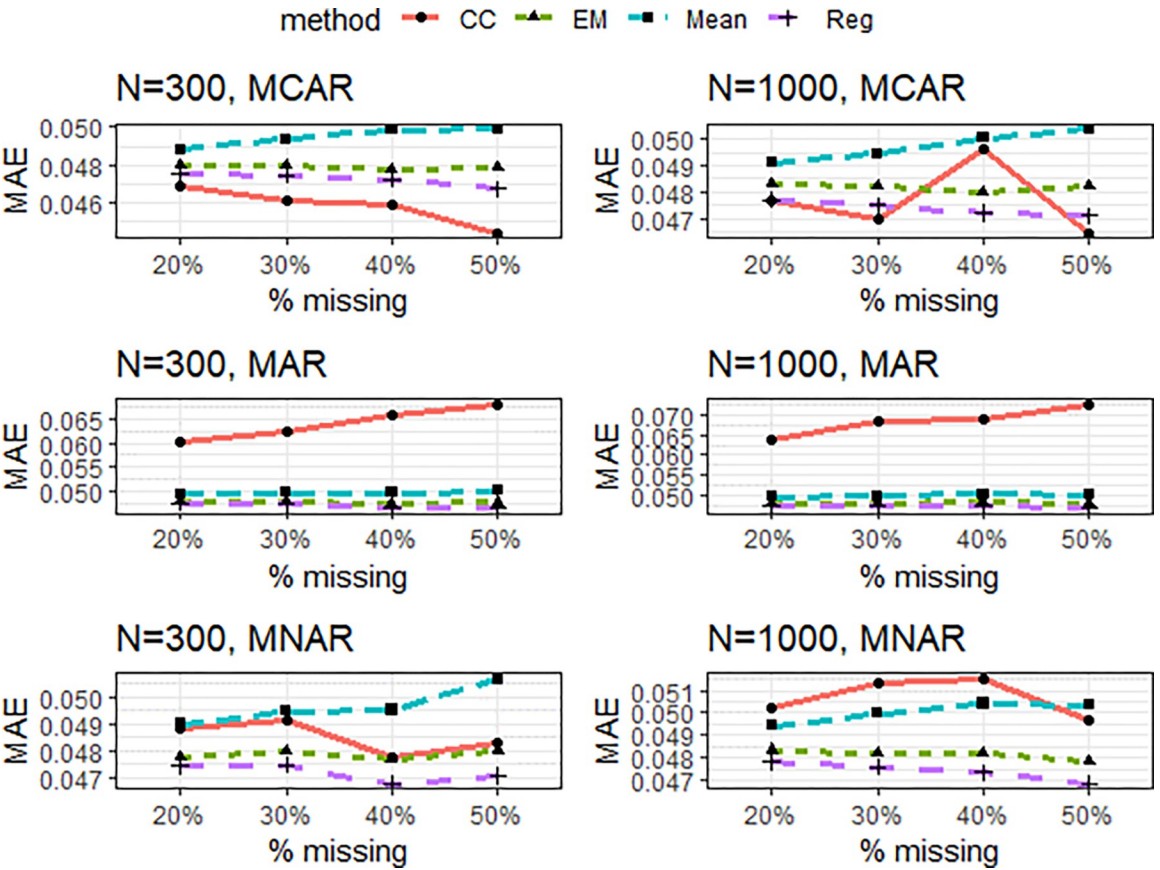

**Fig 7. Simulation example 2: Mean absolute error (MAE) of the different missing data treatments under varying design factor combinations.**

Accordingly, we conducted Monte Carlo simulations to investigate the performance of these single missing data imputation in various conditions that differ in missingness mechanism, missing proportion, and sample size. Two prominent social science models empirically validated by two studies correspondingly were used as benchmarks for our simulations. Our findings are summarized and placed in the context of existing literature where necessary.

In terms of measurement model assessment, none of the methods induced unacceptable values for the reliability and convergent validity measures considered. In addition, the composite reliability and average variance extracted values measuring internal consistency and convergent validity, respectively, were indistinguishable across the board. Though all the considered techniques performed reasonably well in terms of bias in estimating structural parameters, we found that regression imputation outperformed the other candidate methods on average. This finding aligns with a previous study [9]. Complete case analysis had the worst performance except for a few inconsistent best performances under the MCAR mechanism. Across the board, its relative underperformance stood out in assessing the precision of estimates by consistently producing less precise standard errors. These results re-echo what previous critics opined about listwise deletion as being among the worst available missing data treatments for practical applications [30].

For regression and EM imputation techniques, the effect of increasing proportions of missingness was hardly noticeable on the change (increase or decrease) in the bias of estimated structural model parameters. This is similar to what we found in a previous study [12], which

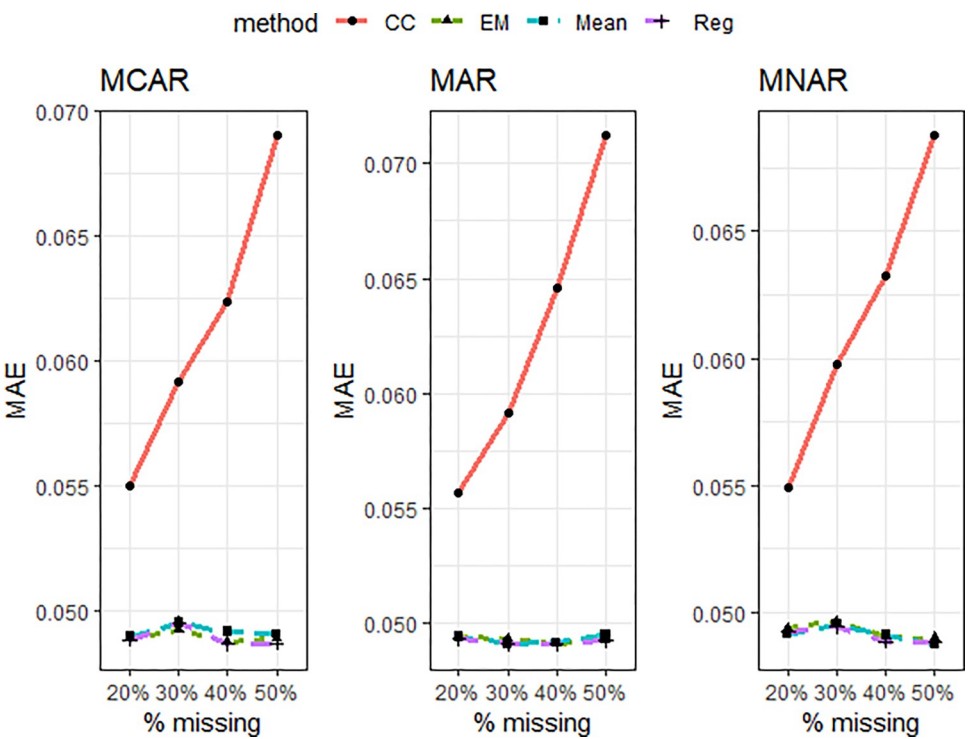

**Fig 8. Standard error (SE) of structural parameter estimates for the different missing data treatments.**

opined that the proportion of missing data, under the MAR assumption, should not guide decision-making in multiple imputations. In most cases, the imputation based on the EM algorithm competed favourably with the regression imputation, especially for the MAR and MNAR mechanisms. This finding is consistent with a study that compared the efficacy of imputation methods for covariance-based structural equation modelling [10]. They found the EM algorithm ranked second or third best, depending on the sample size, in comparing methods like full information maximum likelihood, multiple imputation, EM imputation, mean imputation, and regression imputation [10]. Further, a previous study mentioned EM imputation as one of the promising approaches for treating incomplete data in SEM. They recommended that additional Monte Carlo studies be conducted to assess the utility of the EM algorithm in missing data imputation [11].

This study is not without limitations. First, none of the four imputation approaches considered the uncertainty from missing data in the analysis. This can be problematic, especially when the missingness rate is considerably high, and can result in elevated false positive error rates. Multiple imputations should be used to account for the uncertainty caused by incomplete data. However, no implementation for multiple imputation exists for PLS-SEM in any software or computing programs. Second, it should be noted that the techniques used in this study do not consider the uniqueness of the analysis model and are performed pre-modeling. However, owing to this, they have broad applicability in dealing with missing values. Consequently, imputation techniques linked with the PLS-SEM model should be developed to use model insights fully and, as a result, benefit model estimates. One such method found in the literature, albeit with no software implementation or program accessible, is an EM algorithm-based method the authors termed EM PLS-SEM [31].

We note two previous studies [9,32] that empirically evaluated the impact of incomplete data in PLS-SEM estimation and highlight how this study differs. Both studies restricted the assessment to the MCAR missing data mechanism. [33] investigated solely the listwise deletion approach and focused on the estimation quality of methods other than the PLS-SEM. They compared PLS with maximum likelihood (ML) and full-information maximum likelihood (FIML) methods when estimating SEM. In other words, rather than missing value imputation, which is at the heart of our research, model estimation was the primary focus.

There are limited studies on the assessment of missing data methods for PLS-SEM. In addition, to our knowledge, no previous research has extended the performance evaluation of these missing data techniques for PLS-SEM estimation under different missing data mechanisms. Like any other, our simulation findings may be limited to the scenarios considered by our simulation data. As a result, the findings cannot be applied to situations that have not been investigated. Nevertheless, our results provide some indication of the viability of using single imputation approaches for incomplete data in PLS-SEM.

This research work has quite some theoretical implications. This study is the first to empirically compare missing data methods in PLS-SEM under the different missing data mechanisms. This is notable because of the availability of numerous empirical assessments in the context of the covariance-based SEM, which is relatively higher than PLS-SEM. We cannot make any statements on PLS performance where formative indicators prevail since our theoretical model only includes constructs measured using reflective indicators. Nonetheless, our recommendations should be valuable for practical researchers, among whom there appears to be quite a bit of disparity in the motivations for selecting one approach over another. However, there is no systematic empirical assessment to validate that choice.

## Conclusions

Our simulation experiments reveal the outperformance of the regression imputation against the other alternatives in the recovery of model parameters and precision of parameter estimates. Overall, we found the under-utilized imputation approaches, namely, regression and EM imputation, useful and excellent in performance. They merit more widespread adoption for treating missing values when analyzing PLS-SEM studies.

Findings from this study have far-reaching practical implications for improving data quality, model fit, and decision-making. Properly handling missing data is essential for enhancing the robustness and utility of PLS-SEM in both research and practical applications. While our findings suggest recommending regression imputation over the others, we advise that clearly communicating the chosen missing data treatment and its implication is critical. Practitioners and researchers must convey the limitations and uncertainties associated with these methodologies.Top of Form.

## Supporting information

**S1 File. R scripts used for Monte Carlo simulations and data analyses.**
(ZIP)

## Author Contributions

**Conceptualization:** Lateef Babatunde Amusa.

**Data curation:** Lateef Babatunde Amusa.

**Formal analysis:** Lateef Babatunde Amusa.

**Methodology:** Lateef Babatunde Amusa.

**Project administration:** Twinomurinzi Hossana.

**Software:** Lateef Babatunde Amusa.

**Supervision:** Twinomurinzi Hossana.

**Writing – original draft:** Lateef Babatunde Amusa.

**Writing – review & editing:** Lateef Babatunde Amusa, Twinomurinzi Hossana.

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
