## [Decision Letter · Decision Letter 0]

17 Oct 2023

PONE-D-23-24843An empirical comparison of some missing data treatments in PLS-SEMPLOS ONE

Dear Dr. Amusa,

Thank you for submitting your manuscript to PLOS ONE. After careful consideration, we feel that it has merit but does not fully meet PLOS ONE’s publication criteria as it currently stands. Therefore, we invite you to submit a revised version of the manuscript that addresses the points raised during the review process. When preparing your revised manuscript, you are asked to carefully consider the reviewer comments which are attached, and submit a list of responses to the comments.

We look forward to receiving your revised manuscript.

Kind regards,

Jibril Adewale Bamgbade

Academic Editor

PLOS ONE

Reviewers' comments:

Reviewer's Responses to Questions

**Comments to the Author**

1. Is the manuscript technically sound, and do the data support the conclusions?

Reviewer #1: Yes

2. Has the statistical analysis been performed appropriately and rigorously? 

Reviewer #1: Yes

3. Have the authors made all data underlying the findings in their manuscript fully available?

Reviewer #1: Yes

4. Is the manuscript presented in an intelligible fashion and written in standard English?

Reviewer #1: Yes

5. Review Comments to the Author

Reviewer #1: Thank you for submitting your manuscript to PLOS ONE. I have provided some feedback as detailed below:

Introduction:

The introduction provides a clear background and context, making sense of the pertinence of PLS-SEM and missing data issues. It effectively outlines the research gap and emphasizes the significance of addressing it. Notwithstanding, the introduction is too lengthy and lack compactness. Additionally, while the research gap is mentioned in the introduction, the research objectives and questions could be more clearly stated.

Methodology

The methodology section is well-structured, detailing the approach used for the study. It provides a clear explanation of the missing data mechanisms and design factors considered in the simulations. The use of Monte Carlo simulations is appropriate for this type of study. Nevertheless, the section could be more reader-friendly by breaking down complex technical details into simpler language for non-expert readers. The methodology also lacks a discussion of potential limitations or assumptions made during the simulations.

Discussion

The discussion section effectively summarizes and interprets the results, relating them to the research questions. It compares different missing data imputation techniques, highlighting their strengths and weaknesses. However, the discussion could provide more insight into the practical implications of the findings for researchers using PLS-SEM. It would also benefit from a more comprehensive review of related literature to place the results in a broader context.

Conclusion

The conclusion summarizes the key findings concisely. It reiterates the importance of using regression imputation and EM imputation for addressing missing data in PLS-SEM. The conclusion could however be strengthened by discussing the real-world implications and practical recommendations resulting from the study.

6. PLOS authors have the option to publish the peer review history of their article (what does this mean?). If published, this will include your full peer review and any attached files.

Reviewer #1: No

---

## [Author Response · Author response to Decision Letter 0]

23 Oct 2023

Introduction:

The introduction provides a clear background and context, making sense of the pertinence of PLS-SEM and missing data issues. It effectively outlines the research gap and emphasizes the significance of addressing it. Notwithstanding, the introduction is too lengthy and lack compactness. Additionally, while the research gap is mentioned in the introduction, the research objectives and questions could be more clearly stated.

Response: We have reduced the introduction section by moving the subsection on missing data mechanisms to the methodology section. The introduction section is now more compact. The aim and objectives of the research have now been clearly stated.

Thank you.

Methodology

The methodology section is well-structured, detailing the approach used for the study. It provides a clear explanation of the missing data mechanisms and design factors considered in the simulations. The use of Monte Carlo simulations is appropriate for this type of study. Nevertheless, the section could be more reader-friendly by breaking down complex technical details into simpler language for non-expert readers. The methodology also lacks a discussion of potential limitations or assumptions made during the simulations.

Response: Thank you. We have made assumptions about our simulations that they mimic the social sciences, and by implication, our findings are valid only within the boundaries of the scenarios we investigate. We wrote specifically “We chose this simulation route because it reflects the typical complexity of structural equation models within the social science discipline.” We also wrote in the Discussion “Our simulation findings, like any other, may be limited to the scenarios considered by our simulation data. As a result, the findings cannot be applied to situations that have not been investigated.”

As advised, we have simplified a simplify a few complex technical details. However, most of the other technical terms are standard terms in the statistical literature whose meanings would change if further simplified.

Discussion

The discussion section effectively summarizes and interprets the results, relating them to the research questions. It compares different missing data imputation techniques, highlighting their strengths and weaknesses. However, the discussion could provide more insight into the practical implications of the findings for researchers using PLS-SEM. It would also benefit from a more comprehensive review of related literature to place the results in a broader context.

Response: Thank you, we have provided more insights into the practical implications of the findings. A more comprehensive review of related literature has also been done. 

Conclusion

The conclusion summarizes the key findings concisely. It reiterates the importance of using regression imputation and EM imputation for addressing missing data in PLS-SEM. The conclusion could however be strengthened by discussing the real-world implications and practical recommendations resulting from the study.

Response: Thank you, we have now included the real-world implications and practical recommendations resulting from the study.

---

## [Decision Letter · Decision Letter 1]

28 Dec 2023

An empirical comparison of some missing data treatments in PLS-SEM

PONE-D-23-24843R1

Dear Dr. Amusa,

We’re pleased to inform you that your manuscript has been judged scientifically suitable for publication and will be formally accepted for publication once it meets all outstanding technical requirements.

Kind regards,

Jibril Adewale Bamgbade

Academic Editor

PLOS ONE

Additional Editor Comments (optional):

Reviewers' comments:

Reviewer's Responses to Questions

**Comments to the Author**

1. If the authors have adequately addressed your comments raised in a previous round of review and you feel that this manuscript is now acceptable for publication, you may indicate that here to bypass the “Comments to the Author” section, enter your conflict of interest statement in the “Confidential to Editor” section, and submit your "Accept" recommendation.

Reviewer #1: All comments have been addressed

2. Is the manuscript technically sound, and do the data support the conclusions?

Reviewer #1: Yes

3. Has the statistical analysis been performed appropriately and rigorously? 

Reviewer #1: Yes

4. Have the authors made all data underlying the findings in their manuscript fully available?

Reviewer #1: Yes

5. Is the manuscript presented in an intelligible fashion and written in standard English?

Reviewer #1: Yes

6. Review Comments to the Author

Reviewer #1: Thanks for submitting the corrected manuscript. All the requested comments have been addressed appropriately.

7. PLOS authors have the option to publish the peer review history of their article (what does this mean?). If published, this will include your full peer review and any attached files.

Reviewer #1: No

---

## [Editor Report · Acceptance letter]

10 Jan 2024

PONE-D-23-24843R1 

PLOS ONE

Dear Dr. Amusa, 

I'm pleased to inform you that your manuscript has been deemed suitable for publication in PLOS ONE. Congratulations! Your manuscript is now being handed over to our production team.

Kind regards, 

on behalf of

Dr. Jibril Adewale Bamgbade 

Academic Editor

PLOS ONE